# Technology-Based (Mhealth) and Standard/Traditional Maternal Care for Pregnant Woman: A Systematic Literature Review

**DOI:** 10.3390/healthcare10071287

**Published:** 2022-07-12

**Authors:** Tatik Kusyanti, Firman Fuad Wirakusumah, Fedri Ruluwedrata Rinawan, Abdul Muhith, Ayi Purbasari, Fitriana Mawardi, Indriana Widya Puspitasari, Afina Faza, Alyxia Gita Stellata

**Affiliations:** 1Doctoral Study Program, Faculty of Medicine, Universitas Padjadjaran, Jl. Eyckman No. 38, Bandung 40161, Indonesia; 2Department of Midwifery, Bandung Health Polytechnic, Jl. Sederhana No. 2, Bandung 40161, Indonesia; 3Center of Excellence, Bandung Health Polytechnic, Jl. Pajajaran 56, Bandung 40171, Indonesia; 4Department of Obstetrics and Gynecology, Faculty of Medicine, Universitas Padjadjaran, Jl. Eyckman No. 38, Bandung 40161, Indonesia; firman_fuad@yahoo.com; 5Department of Obstetrics and Gynecology, Dr. Hasan Sadikin General Hospital, Jl. Pasteur No. 38, Bandung 40161, Indonesia; 6Center for Health System Study and Health Workforce Education Innovation, Faculty of Medicine, Universitas Padjadjaran, Jl. Eyckman No. 38, Bandung 40161, Indonesia; f.rinawan@unpad.ac.id; 7Department of Public Health, Faculty of Medicine, Universitas Padjadjaran, Jalan Ir. Soekarno KM. 21, Jatinangor, Sumedang 45363, Indonesia; 8Indonesian Society for Remote Sensing Branch, Gedung Labtek IX-C lt.3 Jalan Ganesha 10, Institut Teknologi Bandung, Bandung 40132, Indonesia; 9Department of Nursing Science, Universitas Nahdlatul Ulama, Surabaya 60237, Indonesia; abdulmuhith@unusa.ac.id; 10Informatics Engineering Study Program, Faculty of Engineering, Universitas Pasundan, Jl. Dr. Setiabudi No. 193, Bandung 40153, Indonesia; pbasari@unpas.ac.id; 11Department of Family and Community Medicine, Faculty of Medicine, Public Health and Nursing, Universitas Gadjah Mada, Yogyakarta 55281, Indonesia; fitriana.8918@ugm.ac.id; 12Master of Midwifery Study Program, Faculty of Medicine, Universitas Padjadjaran, Jl. Eyckman No. 38, Bandung 40161, Indonesia; indriana20001@mail.unpad.ac.id (I.W.P.); alyxia20001@mail.unpad.ac.id (A.G.S.); 13Master of Public Health Study Program, Faculty of Medicine, Universitas Padjadjaran, Jalan Eyckman No. 38 Gedung RSP Unpad Lantai 4, Bandung 40161, Indonesia; afina20005@mail.unpad.ac.id; 14School of Economic and Business, Telkom University, Bandung 40257, Indonesia

**Keywords:** mHealth, standard maternal care, pregnant women

## Abstract

The world of health has changed significantly since the advent of smartphones. Smartphones have been widely known to facilitate the search for health information in the mobile Health (mHealth) system, which is used to improve the quality of life for patients, such as communication between doctors and patients. This systematic literature review aims to identify the use of mHealth as a digital communication tool for pregnant women by comparing technology-based and standard-based pregnancy care. The method used is a systematic review of articles related to pregnancy care that utilize mHealth for pregnant women. The articles were obtained from the database based on the PICO framework; we searched articles using seven databases. The selection was adjusted to the inclusion criteria, data extraction, study quality evaluation, and results from synthesis. From the disbursement, 543 articles were obtained and 10 results were obtained after the screening. After a critical appraisal was carried out, four articles were obtained. Advantages can be in the form of increasing knowledge of pregnant women who use mHealth due to the availability of information needed by pregnant women in the mHealth application. mHealth also provides information about their babies, so the impact of mHealth is not only for mothers. mHealth is a promising solution in pregnancy care compared to the standard of maternal care.

## 1. Introduction

Mobile Health (mHealth) is an essential element of eHealth (electronic Health) [1,2]. mHealth is a term used in health services using smartphones [3]. In the rapidly growing health services era, mHealth must provide optimal service results. mHealth is also very important because it makes health services easier to obtain with the help of communication technology [1]. mHealth is defined by the World Health Organization (WHO) as the use of mobile devices (smartphones) as a monitoring tool for health services [1,3]. The existence of smartphones has changed communication and health systems to be more advanced, along with the increasing demand in the field. Smartphones have been suggested to facilitate search and health information systems. The mHealth system improves the quality of health and patient life, facilitating communication between doctors and patients [4,5,6,7]. However, challenges in its implementation can arise regarding improving and implementing mHealth in the community [8,9]. 

Information and communication technologies, especially the internet, have changed how users search for information and make decisions about health. Among the information and communication technologies that cover the mHealth component as eHealth, health practices are supported by communication tools, such as smartphones, patient monitoring tools, and other wireless devices [10,11]. M-Health technology has developed in lower-middle-income countries [12]. mHealth resources have secure and easy access to care information, including surveillance, health knowledge, teaching, and various research [10]. mHealth application services include communication between users and healthcare systems, such as call centers, appointments, and timely maintenance reminders [1,10].

The health of pregnant women is an important indicator to see the health status of a nation and is one of the components of the development index and quality-of-life index [13]. The international program was initiated by the WHO to provide comprehensive services to the community, knowing that at least 400 million people do not have access to essential health services [14,15]. The health status of pregnant women is influenced by good prenatal care to prevent complications and death during childbirth and fetal growth and health [16,17]. In making visits to health services, it is helpful to identify and know things related to maternal and child health to help reduce maternal mortality (MMR) and infant mortality (IMR) [3,18].

The application can help record information on the health of pregnant women so that pregnant women can obtain information on antenatal care, and health workers can manage examination data appropriately and provide consulting services regarding the information needed through applications that make it easier for pregnant women to overcome the problems they experienced during pregnancy [11,19]. In addition to these problems, the unequal access to health services caused by geographical conditions is not limited; the problem is also related to distance limitations, and so one solution that can be applied in handling pregnant women’s health problems is long-distance health services, often referred to as telehealth, telemedicine, and telenursing [17,20]. Furthermore, telemedicine has been shown to influence economic factors to reduce health costs and increase diagnosis in women with high-risk pregnancies [20].

Pregnant women use their smartphones more often to access information about birth preparation, share experiences, and provide comprehensive support to others on social media [21]. Many articles have discussed using mHealth in pregnancy care. However, there is still limited discussion about comparing pregnancy care using mHealth and conventional pregnancy care in developing countries.

This literature review aims to identify several studies that investigated the use of mHealth as a digital communication tool for pregnant women and compare technology-based and standard-based pregnancy care. This study is expected to improve the health of pregnant women and reduce maternal mortality.

## 2. Materials and Methods

Study design and research strategy. The systematic review included qualitative, quantitative, mixed-method studies, and the concepts contained there were identified using the Population, Intervention, Comparison, Outcome (PICO) framework. A systematic review was proposed through a review of articles related to standard/traditional pregnancy care and those that utilize technology such as mHealth for pregnant women. Reports obtained from the internet are linked to an electronic database. The electronic databases used are Ebsco, Pubmed, Wiley, Springerlink, Sage, BMC, and Proquest.

Article searches were carried out based on the PICO framework: Population: pregnant women in the community; Intervention: telehealth; Comparison: standard care; Outcome: complication prevention. After obtaining articles that complied with the PICO and inclusion criteria, 10 eligible articles were obtained. Then, critical appraisal was carried out using the Critical Appraisal Skills Program (CASP). In addition to CASP, this article uses JBI systematic reviews and the Mixed Methods Appraisal Tool (MMAT) Version 2018.

Article searches were conducted using the keywords “Digital health communication” or “Telemidwifery” or “Telehealth” or “ICTs” and “perception” or “experience” and “maternal” or “pregnant women” and “primary healthcare” or “primary healthcare services”. Article restrictions are carried out using articles published between 2015 and 2021. 

The inclusion criteria in this systematic review are: (1) articles related to pregnancy care use technology; (2) the articles used are those published between 2015 and 2021; (3) original research articles; (4) articles written in English; and (5) articles whose research subjects are pregnant women who have an independent income.

The exclusion criteria for this systematic review are: (1) articles published before 2015; (2) do not use English; (3) incomplete article structure; (4) the article’s subject is not a mother seeking asylum rights in a particular country; (5) literature review systematic articles; and (6) protocol study articles (Figure 1).

## 3. Results

We searched for results using the keywords “Digital health communication” or “Telemidwifery” or “Telehealth” or “ICTs” and “perception” or “experience” and “maternal” or “pregnancy women” and “primary healthcare” or “primary healthcare services”, using electronic databases, including Ebsco, Pubmed, Wiley, SpringerLink, sage, BMC, and Proquest. The search results using these keywords yielded 8 articles from Pubmed, 6 articles from Wiley, 8 articles from Springerlink, 3 articles from SAGE, 16 articles from Proquest, 3 articles from BMC, and 8 articles from EBSCO. We then filtered the articles and obtained 10 articles. Subsequently, the articles were screened based on the full text and the Critical Appraisal assessment obtained four articles (Table 1).

## 4. Discussion

The results of identifying four articles match the search inclusion criteria and the PICO that was determined. The article’s findings revealed the benefits of mHealth in pregnancy care. This discussion will analyze the selected article’s title according to the inclusion criteria and the PICO one by one. In the first article, entitled Mobile health technology for gestational care: evaluation of the GestAção’s app, written by da Silva, Raimunda Magalhães et al., it was found that the level of satisfaction of pregnant women was significant with the use of the terms of mobile application namely: considering the purpose (CVI = 0.92), structure and presentation of the application (CVI = 0.86), and relevance (CVI = 0.92). From these figures, it can be concluded that pregnant women are satisfied with the application’s appearance. This study concluded that the application provides information support and encourages self-care for pregnant women, health promotion activities, and health education tools. The application increases the knowledge of pregnant women and supports consulting activities for them during their pregnancy [13]. This is in line with the research by Murthy, Nirmala, et al. entitled, The Impact of an mHealth Voice Message Service (mMitra) on Infant Care Knowledge. Practices Among Low-Income Women in India: Findings from a Pseudo-Randomized Controlled The trial, said that the impact of the mHealth intervention led to a statistically significant difference in increasing knowledge [7,12]. In the mHealth application, pregnant women can access various information related to their pregnancy so that mothers feel facilitated and satisfied with this mHealth service. Suppose it is associated with Edgar Dale’s theory in the Cone of experience theory. In that case, communication occurs in the learning process, the giver of the material is the communicator, and the material receiver is the communicant. The material presented will be more easily absorbed if using a medium [11].

One of the benefits of mHealth is the media that can be used for digital communication. This system uses information technology that supports remote handling of patient health related to medical personnel. The existence of digital communication can increase the satisfaction of patients (pregnant women) because its use does not require face-to-face contact with medical personnel. The existence of applications that pregnant women use is very useful during their pregnancy. In some of the research results that we have discussed above, it is shown that with these applications, it can be easier to get the information needed during pregnancy. Pregnant women will experience many physical and psychological changes. These changes can be an inconvenience, interfering with the mother during her pregnancy. The inconvenience that occurs, indeed, requires a solution. With information from digital communication, it is possible to detect high-risk pregnancies early so pregnant women can get monitoring to avoid various risks of pregnancy and solutions to solve the problems experienced by mothers. One of the solutions expected by the mother can be presented in educational tools, such as videos, in the application. Media, such as videos, make mothers more interested in listening to information according to the problems they are facing. In addition to providing a solution for the discomfort, the mother can also get information about her pregnancy care according to the gestational age. Therefore, women can expect a healthy and optimal pregnancy outcome.

The article by Murthy, Nirmala et al., above, also stated that the mHealth intervention improves the baby’s health. In this case, the indicator used is the baby’s weight gain. It can be seen that there is an increase in changes in knowledge about infant care and behavior/practices of infant care. The baby’s health may depend on several factors in the above study, including diet, nutritional status, rest rates, and pregnancy complications among women [22]. The mHealth intervention systematically assessed the impact on infant birth weight or demonstrated a positive effect on increasing birth weight. Women exposed to the mMitra message experienced significant improvements in infant care practices, supplementary solid feeding, and infant feeding at six months of age, potentially improving infant growth and nutritional status. There was a statistically significant difference; women exposed to the mMitra message significantly improved behavioral practice. In addition, there is also an increase in knowledge on infant care practices, such as initiation of breastfeeding within one hour after birth, exclusive breastfeeding, complementary foods according to the needs of vitamins and minerals, and adherence to proper nutrition practices, among mHealth users compared to non-users [22,24]. The existence of mHealth for mothers is closely related to the health of the babies they give birth to. One of them is information that mothers easily obtain about various things needed in baby care, specialized care, nutrition, and others. Increasing knowledge from the information obtained by mothers from mHealth certainly requires an application medium that is easy to access and understand and is interesting to use.

The research results conducted by Masoi, Theresia J, and Stephen M. Kibusi, entitled Improving pregnant women’s knowledge on danger signs and birth preparedness practices using an interactive mobile messaging alert system in Dodoma region, Tanzania: a controlled quasi-experimental study, found that there were significant differences [25]. There was a substantial difference between the knowledge of pregnant women who received intervention through SMS warnings in interactive messages and those who followed standard ANC services, as seen from the post-test score in the intervention group, which was higher than the control group. This is due to several things, including the intervention group having the intensity to interact, communicate, and access health education more often than the control group; educational content through interactive SMS alerts provided evidence-based information regularly. In this study, it can be concluded that the warning system on interactive mobile messages has concluded effective in increasing women’s knowledge about danger signs and improving birth preparedness practices [24]. This is in line with research conducted by Chris Smith et al., which aims to evaluate women’s views and experiences in receiving Mobile Technology for Improved Family Planning (MOTIF) interventions, stating that most women have a positive attitude toward telephone-based interventions. Mobile phones support contraceptive use and report it as a convenient way to ask questions or get advice without going to a health center. Although some women find voicemail distracting, the intervention supports contraceptive use by providing information, encouragement, reminders to return to the clinic, reassurance, and suggestions for problems and positively affects the absorption and continuation of contraception. Women reported feeling cared for and receiving support for additional physical and emotional issues. Most women thought that the duration of the intervention and the frequency of messages were acceptable [24]. The existence of an SMS alert system makes it easier for users to remember check-up schedules, danger signs, preparation for childbirth for pregnant women, reminders of contraception schedules, and others. User knowledge was significantly different from the group that did not use the interactive alert system, and the intensity of communication, their interaction, access to health education benefitted users more often than non-users. The educational content should be provided as evidence-based information with regular continuity. Looking at the above phenomenon, we can say that the importance of mHealth for communication interactions has an impact on increasing knowledge due to systematic frequent and easy access.

A multicenter cohort intervention study conducted by Coleman J et al. found that mHealth can increase the intensity of pregnant women in the ANC examinations, with pregnant women completely following the continuity of care services. The SMS Content contains information on pregnant women and maternal psychosocial support, with a reminder to visit ANC; specifically, there is a reminder to acquire vaccinations for newborns. This study demonstrates the positive impact of mHealth with an increase in achieving the complete mother–infant care continuum [18]. A study developed the eHealth Babies application as an Android-based smartphone application for pregnant women hospitalized in a tertiary hospital in a low-socioeconomic community, intending to provide health information about early pregnancy to increase maternal confidence and reduce anxiety [23]. The existence of mHealth from this research dramatically contributes to the continuity of care. The importance of continuity of care in determining outcomes during pregnancy is closely related with continuity of care. It is hoped that all developments during pregnancy care can be monitored optimally and early detection of abnormalities during pregnancy can also run optimally. In mHealth, it is closely related to an increase in the intensity of the examination (reminder) and maternal psychosocial support during pregnancy care. This social support can increase the comfort and self-confidence of the mother during pregnancy. The existence of mHealth, in this case, really supports a healthy pregnancy and optimal results.

A total of three journal articles have benefits with an emphasis on increasing the knowledge of pregnant women and benefits regarding continuity of care (Continuity of Care) success with a reminder in the application so that it affects service outcomes during pregnancy [7,12,25]. Based on a journal article by Silva et al., it was found that the benefits affect the mother’s knowledge because it can make the mother feel satisfied in using the application. The application can help pregnant women increase knowledge about pregnancy and is helpful during consultations and care for pregnant women [23,25]. Meanwhile, according to Murthy et al., the benefits of knowledge gained by mothers are also due to the mHealth intervention that affects the care practices carried out by mothers [7]. Furthermore, the benefits of knowledge are also obtained by mothers because of the reminder message function from the application used based on research conducted in Tanzania [25]. In addition, the benefits of knowledge can be felt because of an increase in getting pregnancy healthcare on a continuum, and it is due to the use of applications by mothers [23]. Then, the benefits of increasing knowledge of pregnant women are also felt because pregnant women become more focused on applications devoted only to the fetus and pregnancy [26]. The increased knowledge of pregnant women, which is one of the perceived benefits, is also due to efforts to provide correct information and correct bias/wrong information circulating in the community [27].

In addition to the benefits of increasing maternal knowledge, several articles also show a close relationship between the knowledge and behavior of pregnant women and their environment. Among them, one is enabling mothers to adopt better ways of caring for newborns and helping people around the mother play a better role in her pregnancy. Another is allowing the mothers to adopt better practices of caring for newborns and encouraging people around the mother to play a better position in her pregnancy [28]. The benefit of a change in behavior from the application used is a change in public trust, which initially deviated and believed in the myths of pregnancy circulating in the community to become scientifically based beliefs that were more logical and correct [29]. Based on the perceived behavioral benefits, in one study in India, the mHealth Intervention (M-SAKHI) was established to effectively reduce stunting conditions in rural India using the mHealth application to improve the nutritional health condition of pregnant women and their babies, thus, increasing the behavior of pregnant women to seek information related to their pregnancy care, improving personal hygiene during pregnancy, and avoiding the incidence of disease during pregnancy [30]. A study developed the eHealth Babies application as an Android-based smartphone application for pregnant women in a tertiary hospital in a low-socioeconomic community, intending to provide health information about early pregnancy to increase maternal confidence and reduce anxiety [23,26]. In addition to the benefits of mHealth above, it was also obtained from the results of research conducted by Begum T et al. that this is in line with research that states that mHealth can improve the timeliness and completeness of data reporting over time. Overall, understanding of family passive sensing and family awareness of the potential benefits to mother and baby are the main modifiable factors that increase acceptance and reduce gaps in data collection [31].

MHealth is very important and valuable during pregnancy care, but we cannot deny that some obstacles will be encountered. One of them is geographic location. Pregnant women who live in remote geographic areas lack transportation, a worker who already has a previous child faces significant barriers to attending in-person visits, affecting their ability to access this pregnancy care. On the other hand, several studies have shown that telemedicine produces similar outcomes, increases patient satisfaction for low-risk pregnancies, and increases access to subspecialty care for those living in underserved areas. Besides that, low education and lack of digital literacy will be obstacles to the effective use of mHealth in pregnancy care. The problem with mHealth is that not all people can use it properly. Educational-level factors, lack of supporting infrastructure, and local government policy factors can be the obstacles in using mHealth in pregnancy care.

## 5. Conclusions

From the results of the 10 selected articles, a critical appraisal was carried out, and 4 articles were obtained, including various studies according to the Critical Appraisal instrument and in-depth research. Information obtained from reviewed journals shows that pregnancy care using mHealth has many advantages over conventional/standard pregnancy care. This advantage can be in the form of increasing the knowledge of pregnant women who use mHealth due to the availability of information needed by pregnant women in the mHealth application. In addition to providing the information required by pregnant women during their pregnancy about everything needed by pregnant women, mHealth can also give information on postnatal care for mothers and their babies so that the impact of mHealth is not only for mothers in the form of changes in knowledge, but which ultimately affects changes in behavior. However, it also affects the baby’s health, in various aspects, one of which can be seen in the increase in the baby’s weight. In this case, mHealth can also improve the baby’s health, not only the health of pregnant women. Furthermore, mHealth can also provide a reminder service that impacts increasing the regularity of pregnant women in checking their pregnancies and implementing continuity of care in their pregnancy care. This can happen due to the high intensity of mHealth interactions with mothers in the form of reminders, making it easier for mothers to do things according to their needs during pregnancy. 

In general, mHealth and Telehealth for pregnant women, especially during the pandemic, are very necessary. Pregnancy care using mHealth differs from pregnancy care using traditional systems. Traditional pregnancy care is standard/routine pregnancy care that is carried out without advanced technology, such as consulting services using smartphones, chatbots, digital communications, video consultations with smartphones, teleconferences, and video education on smartphones related to pregnancy, birth, and baby care. Moreover, other information is needed during the perinatal period. Traditional/standard/routine maternity care is without the use of media and relies on face-to-face meetings for consultation and monitoring. 

There is great importance in using mHealth, one reason being the pandemic. Pregnant women have unique, different, and varied needs, especially during the COVID-19 pandemic. The field of midwifery care needs to develop a service system for pregnant women that minimizes the exposure of pregnant women to sick people. To reduce the potential risk of face-to-face antenatal visits, telemedicine intelligence is needed to help maintain the continuity of pregnancy care. mHealth also improves the output of pregnant women to delivery with optimal health outcomes for mothers and babies; in this case, pregnant women and babies are becoming the targets for optimal service outcomes, so it is hoped that an intelligent generation will grow.

## 6. Implications 

Implications for practitioners. mHealth uses real-time communication between the patient and practitioner, which has emerged as an effective and valuable treatment modality. mHealth is a rapidly growing healthcare field, increasing patient access to care, reducing costs, improving patient satisfaction, reducing travel time, facilitating complex quality care for patients in rural locations, and increasing efficiency. For practitioners who deal with patients in various settings, mHealth can help improve healthcare delivery to underserved populations. Electronic information and telecommunication technologies can be used to support long-distance clinical healthcare, a growing trend that offers improved patient access and more engaged patients with better outcomes. For midwives who provide direct healthcare to patients (pregnant women) in rural and urban settings and locations, ranging from clinics and hospitals to emergency/urgent care sites, nurse/midwife practitioner practices, and nursing homes, among others, using mHealth can help improve the provision of healthcare to underserved pregnant women. 

For academics, the implications of mHealth in healthcare are to innovate the use of technology in health in the form of early detection, routine remote care, treatment, monitoring, and evaluation of customer satisfaction implemented by mHealth. Therefore, the evaluation of efficient customer satisfaction and the effectiveness of mHealth utilization and evaluation of mHealth needs for various cases will be a consideration for academics to create the latest innovations, following evidence-based information, so that the mHealth created will develop optimally according to needs and current circumstances.

## Figures and Tables

**Figure 1 healthcare-10-01287-f001:**
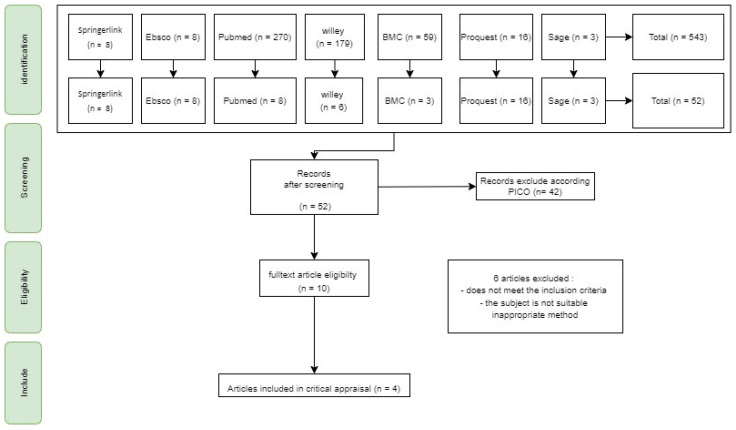
Prisma Flowchart.

**Table 1 healthcare-10-01287-t001:** Article Table. The 4 articles are as follows below.

No	Writer/Title	Country	Year	Sample	Result	Critical Appraisal Result
1	da Silva, Raimunda Magalhaes et al. Mobile health technology for gestational care: evaluation of the GestAção’s app [12]	Brazil	2019	Pregnant woman	Intervention: Use of GestAção’s app for pregnant women Findings: studies prove a significant level of satisfaction of pregnant women with the use of the application, considering that purpose (CVI = 0.92), structure and presentation of the application (CVI = 0.86), and relevance (CVI = 0.92). The application also provides information support and encourages self-care for pregnant women and health promotion activities as well as health education tools. This increases the knowledge of pregnant women and supports consultation activities for pregnant women	included
2	Murthy, Nirmala et al. The Impact of an *mHealth* Voice Message Service (mMitra) on Infant Care Knowledge, and Practices Among Low-Income Women in India: Findings from a Pseudo-Randomized Controlled Trial [8]	India	2019	Pregnant Woman	In this study add to the growing evidence on the impact of the *mHealth* intervention with statistically significant differences in knowledge and behavior that are known to improve infant and infant health outcomes, in this case in terms of infant birth weight. It can be concluded that the impact of M Health on this research are: Improve children’s health Increase mother’s knowledge	included
3	Masoi, Theresia J dan Stephen M. Kibusi Improving pregnant women’s knowledge on danger signs and birth preparedness practices using an interactive mobile messaging alert system in Dodoma region, Tanzania: a controlled quasi experimental study [22]	Tanzania	2019	Pregnant woman	This journal has compared the knowledge of pregnant women who received interventions using “The interactive SMS alert system” in the form of interactive messages about danger signs and childbirth readiness “where the control group is pregnant women who receive standard ANC.Result: There is a significant difference between the knowledge of pregnant women who receive intervention through SMS warnings in the form of interactive messages and those who follow standard ANC services as seen from the post test score in the intervention group which is higher than the control group.	included
4	Coleman J et al. Evaluating the effect of maternal *mHealth* text messages on uptake of maternal and child healthcare services in South Africa: a multicentre cohort intervention study [23]	South Africa	2020	Pregnant Woman	Intervention: SMS in MAMA Control: MAMA Result: Increase the intensity of pregnant women in ANC examination Pregnant women are more complete in participating in continuum of care services	Included

## Data Availability

Not applicable.

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
