# Peer review of "Technology-Based (Mhealth) and Standard/Traditional Maternal Care for Pregnant Woman: A Systematic Literature Review"

_healthcare, 2022, doi:10.3390/healthcare10071287_

Round 1

Reviewer 1 Report

The discussed problem of eHealth is crucial and has a meaningful impact on perinatal care, especially in covid- and post covid times. Moreover, in the areas where medical providers are less available, providing the app with medical care seems reasonable. 

The introduction section is long and chaotic without a typical funnel model. It could be written a little bit shorter and more precise about the problem discussed in the study. Moreover, the aim of the study is hidden from the reader. I suppose to make the study's objective separate in the last paragraph of the introduction.

The methodology is the weakest part of the study, which does not make it possible to repeat the research. There is very little information on how the analysis was performed. The criteria for including studies are entirely unclear. The methodology does not look like a systematic review was conducted. I suppose to change the name of the study to “narrative review” or perform the analysis once again using PRISMA guidelines for the systematic reviews, which you can find on the EQUATOR website, or you can compare your study to the article Feduniw et al. doi: 10.1097/MD.0000000000023681.

Line 135-139 PICO question is made unclear. Could you explain what you mean by “maternal in the community”? It seems not entirely correct. Do you mean mothers with children in the community? About which community was written in the manuscript? “Intervention: telemedicine”, please specify it. Etc. Setting, assessing of biases, selection process description is missed.

Databases were screened to make an impression “Ebsco, Pubmed, Whitley, Sprigerlink, Sage, Cochrane, Google Scholar, BMC, ProquestNevertheless, the BMC, Whitley, and Sprigerlink are not databases, and all of them are indexed in PubMed/Medline. Therefore, mentioning these particular names makes little sense for me. Databases which are missing are:  Embase, Web of Science and Scopus. The revision of grey literature like Google Scholar is very admiring. How many articles do the search show? The flow diagram is not readable, and I do not believe it. 

Using the Search strategy: [Digital health communication OR Telemidwivery OR Tele-Health OR ICTs AND perception OR (experience AND maternal) OR (pregnancy women AND primary health care) OR primary health care services] and limitation of 2015-2021 on PubMed gives 89,547 results (at 21.05.2022) and on Google Scholar 24 800 results (at 21.05.2022) what is entirely different to results of the study (total 551). The study flow diagram was also performed inappropriately. I do not believe there were no duplicates in such similar analyzed databases. It’s not sure why the authors have screened the years 2015-2021. Please perform the analysis regardless of time or explain why this restriction was used.

The Result section is missing in the study. Table 1 and its repetition in the text are no results. It is only a short description of 4 studies, particularly its results. I do not believe that any studies assessing eHealth tools were performed in USA or Europe, where smartphones are more commonly used. 

Author Response

Here I include input and answers from the revised results :

The discussed problem of eHealth is crucial and has a meaningful impact on perinatal care, especially in covid- and post covid times. Moreover, in the areas where medical providers are less available, providing the app with medical care seems reasonable. 

Response : Thank you very much for your feedback.

Editor Comments :

  1. The introduction section is long and chaotic without a typical funnel model. It could be written a little bit shorter and more precise about the problem discussed in the study. Moreover, the aim of the study is hidden from the reader. I suppose to make the study's objective separate in the last paragraph of the introduction

Response : thank you for the feedback, We've revised the introduction to be more concise and condensed and separated the objectives into separate paragraphs.

  1. The methodology is the weakest part of the study, which does not make it possible to repeat the research. There is very little information on how the analysis was performed. The criteria for including studies are entirely unclear. The methodology does not look like a systematic review was conducted. I suppose to change the name of the study to “narrative review” or perform the analysis once again using PRISMA guidelines for the systematic reviews, which you can find on the EQUATOR website, or you can compare your study to the article Feduniw et al. doi: 10.1097/MD.0000000000023681.

Response : thank you for your feedback, We have added information about the methodology to the critical assessment section of the selected articles. 

  1. Line 135-139 PICO question is made unclear. Could you explain what you mean by “maternal in the community”? It seems not entirely correct. Do you mean mothers with children in the community? About which community was written in the manuscript? “Intervention: telemedicine”, please specify it. Etc. Setting, assessing of biases, selection process description is missed.

Response : thank you for your feedback. We have explained the meaning of lines 135-139 in the methodology section.

  1. Databases were screened to make an impression “Ebsco, Pubmed, Whitley, Sprigerlink, Sage, Cochrane, Google Scholar, BMC, Proquest”. Nevertheless, the BMC, Whitley, and Sprigerlink are not databases, and all of them are indexed in PubMed/Medline. Therefore, mentioning these particular names makes little sense for me. Databases which are missing are:  Embase, Web of Science and Scopus. The revision of grey literature like Google Scholar is very admiring. How many articles do the search show? The flow diagram is not readable, and I do not believe it. 

Response : thank you for your feedback. We have revised the database and have readjusted it to the flowchart that has been created. whitley in this manuscript means wiley. Sorry for the mistakes in our writing

  1. Using the Search strategy: [Digital health communication OR Telemidwivery OR Tele-Health OR ICTs AND perception OR (experience AND maternal) OR (pregnancy women AND primary health care) OR primary health care services] and limitation of 2015-2021 on PubMed gives 89,547 results (at 21.05.2022) and on Google Scholar 24 800 results (at 21.05.2022) what is entirely different to results of the study (total 551). The study flow diagram was also performed inappropriately. I do not believe there were no duplicates in such similar analyzed databases. It’s not sure why the authors have screened the years 2015-2021. Please perform the analysis regardless of time or explain why this restriction was used.

Response : thank you for your feedback. After we discussed with the writing team regarding the results of searching for articles in the database, we found that:

  1. Team 1
  • Browse articles by compiling a PICO first. Population : pregnant woman in community, Intervention: use digital communication, Comparison: standard care, Outcome: complication prevention.
  • Arrange keywords using the boolean method, namely Key word : Maternal in community AND telemedicine or telehealth or e-health or m-health AND complications or problems or challenges AND prevention or intervention or treatment or program
  • First Search ; Ebsco : 4,015.642 , to 9 filtered through, Full text, Year 2016-2021, Pregnancy, Source type : Academic journals, Major headings: pregnancy complications and telemedicine
  • Springerlink : 257, after filtering through 2016-2021, english language, academic articles, open access, discipline medicine & public health, subdiscipline maternal and child health, reproductive health, gynecology, public health, pediatrics
  • Scanned by title, Ebsco selected 8 (there is 1 duplicate), Springerlink : 8
  • Scanned through the abstract, Selected ebsco : 5, Springerlink : 3
  • In critical appraisal and synthesis, Ebsco : 0, Springerlink : 2 (the article is ; (1)The Impact of an m-Health Voice Message Service (mMitra) on Infant Care Knowledge, and Practices Among Low-Income Women in India: Findings from a Pseudo-Randomized Controlled Trial ; (2) Improving pregnant women’s knowledge on danger signs and birth preparedness practices using an interactive mobile messaging alert system in Dodoma region, Tanzania: a controlled quasi experimental study)
  1. Team 2
  • search in sage, proquest and hindawi. Entering according to keywords can get as many results, Proquests: 2,683,949, Sage:1.418, hindawi: 0.
  • then filtered by year, language, type of literature more conical we get: Proquest: 195.555, Sage:32
  • Then it is downloaded, after that it is tracked according to the inclusion criteria, it is obtained: Proquests: 16, Sage:3
  • (From what was obtained, a critical appraisal was carried out and only 1 article was included, namely with the title : Evaluating the effect of maternal m-Health text messages on uptake of maternal and child health care services in South Af-rica: a multi-centre cohort intervention study.)
  1. Team 3
  • based on the search results in pubmed found 270 filtered articles and after applying the inclusion criteria, 8 articles were found.
  • Based on a search in Wiley, 179 articles were filtered and after applying the inclusion criteria, 6 articles were found.
  • based on a search in BMC, 59 articles were filtered and after checking the inclusion criteria, 3 articles were obtained.
  • (From what was obtained, a critical appraisal was carried out and only 1 article was included, namely with the title : Mobile health technology for gestational care: evaluation of the GestAção’s app)

About a search on google scholar with 8 results, sorry we had an error. The screening results from pubmed should have been 8 results, and apparently no one from the team looked for it on google scholar. sorry for our omission in writing.

  1. The Result section is missing in the study. Table 1 and its repetition in the text are no results. It is only a short description of 4 studies, particularly its results. I do not believe that any studies assessing eHealth tools were performed in USA or Europe, where smartphones are more commonly used. 

Response : thank you for your feedback. We have added and deepened in the results section, please input if there are still errors

Best Regards

Tatik Kusyanti

Reviewer 2 Report

Dear Author,

This research paper is not very new as we have past works related to the field; nevertheless, the context of research is under-explored. The paper can offer an inclusive insight pertaining to such context. Literature review is adequate and cites an appropriate range of literature sources. Methodology is clear; besides, adequate analyses have been carried out with the research objectives. Findings are presented in a good manner. However, the research has not rigor theoretical and managerial implications. Thus, the paper should contain several separate sections that describe implications for academics and practitioners appropriately. The paper needs some improvements in order to meet the standards of this journal.

Author Response

Tatik Kusyanti

Faculty of Medicine

 Padjajaran University

May 11th, 2022

Dear Editor;

I refer the manuscript entitled “TECHNOLOGY BASED (m-Health) AND STANDART/ TRA-DITIONAL MATERNAL CARE ON PREGNANT WOMAN: SYSTEMATIC LITERATURE REVIEW” to be considered for publication in Healthcare. The reason for sending your that the world of Health requires breakthroughs by utilizing digital technology to achieve the goals of health services to the fullest.

Here I include input and answers from the revised results :

This research paper is not very new as we have past works related to the field; nevertheless, the context of research is under-explored. The paper can offer an inclusive insight pertaining to such context. Literature review is adequate and cites an appropriate range of literature sources. Methodology is clear; besides, adequate analyses have been carried out with the research objectives. Findings are presented in a good manner. However, the research has not rigor theoretical and managerial implications. Thus, the paper should contain several separate sections that describe implications for academics and practitioners appropriately. The paper needs some improvements in order to meet the standards of this journal.

Response :. Thank you for your feedback and opinion, we are very happy to hear it. Suggested additions have been added to the article. Hopefully it can meet the expectations of the reviewer and if there are shortcomings, we ask for input on our articles to make them better.

Through this document, MDPI assumes exclusive rights to edit, publish, reproduce, distribute copies, prepare derivative works on paper, electronic or multimedia and include the article in national and international indexes and bibliographic databases.

Thanking you in advance for your cooperation and awaiting your prompt response;

Sincerely,

Tatik Kusyanti

Reviewer 3 Report

In this manuscript by Kusyanti et al, the authors have basically gathered information from 4 articles and elaborated them. However, the input from the authors about the presentation of their review is missing. What are the advantages and disadvantages of telehealth and what the review itself says about it? Major content of the manuscript describes the article by Murthy et al, whereas the critical analysis, ideas and scientific/statistical interpretation of e-health/mobile health concept are not presented. What are the inclusion and exclusion criteria of selecting 4 articles described in the manuscript? Line 150-151: How many articles filtered?

Author Response

Tatik Kusyanti

Faculty of Medicine

 Padjajaran University

May 11th, 2022

Dear Editor;

I refer the manuscript entitled “TECHNOLOGY BASED (m-Health) AND STANDART/ TRA-DITIONAL MATERNAL CARE ON PREGNANT WOMAN: SYSTEMATIC LITERATURE REVIEW” to be considered for publication in Healthcare. The reason for sending your that the world of Health requires breakthroughs by utilizing digital technology to achieve the goals of health services to the fullest.

Here I include input and answers from the revised results :

articles and elaborated them. However, the input from the authors about the presentation of their review is missing. What are the advantages and disadvantages of telehealth and what the review itself says about it? Major content of the manuscript describes the article by Murthy et al, whereas the critical analysis, ideas and scientific/statistical interpretation of e-health/mobile health concept are not presented. What are the inclusion and exclusion criteria of selecting 4 articles described in the manuscript? Line 150-151: How many articles filtered?

Response : thank you for the feedback that has been given. advantages and reviews about e-health have been added to the discussion section. inclusion and exclusion criteria have been added. The process of selecting 4 articles from search to critical appraisal has been added.

After we discussed with the writing team regarding the results of searching for articles in the database, we found that:

  1. Team 1
  • Browse articles by compiling a PICO first. Population : pregnant woman in community, Intervention: use digital communication, Comparison: standard care, Outcome: complication prevention.
  • Arrange keywords using the boolean method, namely Key word : Maternal in community AND telemedicine or telehealth or e-health or m-health AND complications or problems or challenges AND prevention or intervention or treatment or program
  • First Search ; Ebsco : 4,015.642 , to 9 filtered through, Full text, Year 2016-2021, Pregnancy, Source type : Academic journals, Major headings: pregnancy complications and telemedicine
  • Springerlink : 257, after filtering through 2016-2021, english language, academic articles, open access, discipline medicine & public health, subdiscipline maternal and child health, reproductive health, gynecology, public health, pediatrics
  • Scanned by title, Ebsco selected 8 (there is 1 duplicate), Springerlink : 8
  • Scanned through the abstract, Selected ebsco : 5, Springerlink : 3
  • In critical appraisal and synthesis, Ebsco : 0, Springerlink : 2 (the article is ; (1)The Impact of an m-Health Voice Message Service (mMitra) on Infant Care Knowledge, and Practices Among Low-Income Women in India: Findings from a Pseudo-Randomized Controlled Trial ; (2) Improving pregnant women’s knowledge on danger signs and birth preparedness practices using an interactive mobile messaging alert system in Dodoma region, Tanzania: a controlled quasi experimental study)
  1. Team 2
  • search in sage, proquest and hindawi. Entering according to keywords can get as many results, Proquests: 2,683,949, Sage:1.418, hindawi: 0.
  • then filtered by year, language, type of literature more conical we get: Proquest: 195.555, Sage:32
  • Then it is downloaded, after that it is tracked according to the inclusion criteria, it is obtained: Proquests: 16, Sage:3
  • (From what was obtained, a critical appraisal was carried out and only 1 article was included, namely with the title : Evaluating the effect of maternal m-Health text messages on uptake of maternal and child health care services in South Af-rica: a multi-centre cohort intervention study.)
  1. Team 3
  • based on the search results in pubmed found 270 filtered articles and after applying the inclusion criteria, 8 articles were found.
  • Based on a search in Wiley, 179 articles were filtered and after applying the inclusion criteria, 6 articles were found.
  • based on a search in BMC, 59 articles were filtered and after checking the inclusion criteria, 3 articles were obtained.
  • (From what was obtained, a critical appraisal was carried out and only 1 article was included, namely with the title : Mobile health technology for gestational care: evaluation of the GestAção’s app)

Through this document, MDPI assumes exclusive rights to edit, publish, reproduce, distribute copies, prepare derivative works on paper, electronic or multimedia and include the article in national and international indexes and bibliographic databases.

Thanking you in advance for your cooperation and awaiting your prompt response;

Sincerely,

Tatik Kusyanti

Reviewer 4 Report

Thank you for the opportunity to review this article. This paper describes a systematic review of technology Based (mHealth) and standard/ traditional maternal care for pregnant women. I found the article interesting, and I think that it can be helpful for other researchers, as I believe that it offers essential information. Nevertheless, some minor aspects could be revised. These are the following:

- There are some typos in the title.

- The Materials and Methods section describes the methods adequately, but I believe some aspects could be described more precisely. For example, the selection process of the article, described at the beginning of the results, perhaps could fit better in the methods section. This part could also be better explained.

- I found the discussion very interesting. I would propose the authors include. Nevertheless, there are few evaluations of the eHealth or mHealth interventions performed from the users' point of view, as stated in https://doi.org/10.1016/j.cmpb.2021.106462. I would invite the authors to mention this important aspect in the discussion, as I believe that it could be helpful to know the authors’ point of view about this aspect.

Author Response

Tatik Kusyanti

Faculty of Medicine

 Padjajaran University

May 11th, 2022

Dear Editor;

I refer the manuscript entitled “TECHNOLOGY BASED (m-Health) AND STANDART/ TRA-DITIONAL MATERNAL CARE ON PREGNANT WOMAN: SYSTEMATIC LITERATURE REVIEW” to be considered for publication in Healthcare. The reason for sending your that the world of Health requires breakthroughs by utilizing digital technology to achieve the goals of health services to the fullest.

Here I include input and answers from the revised results :

Thank you for the opportunity to review this article.

This paper describes a systematic review of technology Based (mHealth) and standard/ traditional maternal care for pregnant women. I found the article interesting, and I think that it can be helpful for other researchers, as I believe that it offers essential information. Nevertheless, some minor aspects could be revised. These are the following:

  1. There are some typos in the title.

         Response: thank you for your feedback, Sorry for the error in writing the                words in the title and we have corrected it

  1. The Materials and Methods section describes the methods adequately, but I believe some aspects could be described more precisely. For example, the selection process of the article, described at the beginning of the results, perhaps could fit better in the methods section. This part could also be better explained.

         Response : thank you for the feedback you have given. ok we will explain             the process of selecting articles in the method section

  1. I found the discussion very interesting. I would propose the authors include. Nevertheless, there are few evaluations of the eHealth or mHealth interventions performed from the users' point of view, as stated in https://doi.org/10.1016/j.cmpb.2021.106462. I would invite the authors to mention this important aspect in the discussion, as I believe that it could be helpful to know the authors’ point of view about this aspect.

          Response : thank you for the feedback you have given. we have read the              article that you have suggested before. related to our article, we can add              the following:

         evaluation of e-health/m-health interventions from the user's point of                  view    has various very important benefits. These benefits can be                         classified   as follows:

  1. usability

       e-health is very important in terms of increasing knowledge about health             care, one of which is for pregnant women, e-health also provides                         information about anticipating danger signs. In addition, e-health can also           be a reminder in the schedule of pregnancy care visits.

  1. trust and confidence

        e-health provides information from pregnancy to care for the baby, so that           it  has a positive effect on the psychology of the mother and baby which             indirectly grows the mother's confidence in taking care of the baby.

  1. accessibility

       With the existence of e-health, it is easier for users to get access to the                  information needed according to the conditions they are experiencing.                This   is very useful, especially for mothers who are in geographic areas                 that    are difficult to reach and meet face-to-face with health workers.

Through this document, MDPI assumes exclusive rights to edit, publish, reproduce, distribute copies, prepare derivative works on paper, electronic or multimedia and include the article in national and international indexes and bibliographic databases.

Thanking you in advance for your cooperation and awaiting your prompt response;

Sincerely,

Tatik Kusyanti

Round 2

Reviewer 2 Report

Dear Author,

Once again, this research paper is not very new as we have past works related to the field; nevertheless, the context of research is under-explored. The paper can offer an inclusive insight pertaining to such context. Literature review is adequate and cites an appropriate range of literature sources. Methodology is clear; besides, adequate analyses have been carried out with the research objectives. Findings are presented in a good manner. However, the research has not rigor theoretical and managerial implications. Thus, the paper should contain several separate sections that describe implications for academics and practitioners appropriately. The paper needs some improvements in order to meet the standards of this journal.

Response : Implications for practitioners, m-Health use real-time communication between patient and practitioner, has emerged as an effective and valuable treatment modality. M-Health is a rapidly growing field of healthcare, increasing patient access to care, reducing costs, improve patient satisfaction, reduce travel time, facilitate complex quality care for patients in rural locations and increasing efficiency. Practitioners who deal with patients in a variety of settings, m-Health can help improve healthcare delivery to underserved populations. uses electronic information and telecommunications technologies to support long-distance clinical healthcare, a growing trend that offers improved patient access, cost savings, and more engaged patients with better outcomes. for midwife who provide direct healthcare to patients (pregnant women) in rural and urban settings and in locations ranging from clinics and hospitals to emergency/urgent care sites, nurse/midwife practitioner practices, and nursing homes, among others, using m-Health can help improve provision of health care to underserved pregnant women.

For academics the implications of m-Health in health care are to innovate the use of technology in health, in the form of early detection, routine remote care, treatment, monitoring and evaluation of customer satisfaction implemented by m-health. The evaluation of efficient customer satisfaction and the effectiveness of m-health utilization, evaluation of m-health needs for various different cases will be a consideration for academics to create the latest innovations in accordance with evidence based so that the m-health created will develop optimally according to the needs, current circumstances.

Reviewer 3 Report

The authors have somehow improved the manuscript. The authors should present rigorous and exhaustive comparison and analysis of telehealth/M-Health to that of traditional maternal care. To be very specific, what exactly do authors mean by traditional care? How impactful is telehealth in regards to geographical location (rural/sub-urban/urban) and educational background? In general, why the authors emphasize telehealth is important for pregnant women, but not for other terrible diseases.

Response :

Pregnancy care using m-Health is different with the traditional pregnancy care. Traditional pregnancy care is a standard/routine pregnancy care that is carried out without utilizing advanced technology such as consulting services using smartphones, chatbots, digital communications, video consultations with smartphones, teleconferences and video education on smartphones related to pregnancy, birth and baby care, and other information that needed during the perinatal period. Traditional/standard/routine maternity care is  not using the media as mentioned above and relying on face-to-face meetings for consultation and monitoring.

M-health actually is very important and useful during pregnancy care, but we cannot deny that some obstacles will definitely be encountered. One of them is  geographic location. Pregnant women who lived in remote geographic area a lack of transportation, a worker and already having a previous child facing a significant barriers to attend in-person visits, affecting their ability to access this pregnancy care. Several studies have shown that telemedicine produces similar outcomes and increases patient satisfaction for low-risk pregnancies and increases access to subspecialty care for those living in underserved areas. Besides that, low education and lack of digital literacy will be an obstacle to the effective use of m-Health in pregnancy care. The problem when using telemedicine is that not all of them can  using this properlythis can be caused by educational factors of the health care system, such as Lack of supporting infrastructure may play an important role in this regard. In addition, there are local government policy factors that can be an obstacle to use m-Health in pregnancy care.

In general, m-Health, Tele-Health for pregnant women, especially during the pandemic is very necessary.

Pregnant women have unique, different and varied needs, especially during the COVID-19 pandemic. The field of midwifery care needs to develop a service system for pregnant women that minimizes the exposure of pregnant women with sick people. To reduce the potential risk of face-to-face antenatal visits, telemedicine intelligence is needed to help maintain continuity of pregnancy care. Telemedicine also improve the output of pregnant women to delivery with optimal health outcomes for mothers and babies, while in this case pregnant women and babies are become the targets for optimal service outcomes so that it is hoped that a smart generation will grow.